# Clinical Effect of Lenvatinib Re-Administration after Transcatheter Arterial Chemoembolization in Patients with Intermediate Stage Hepatocellular Carcinoma

**DOI:** 10.3390/cancers14246139

**Published:** 2022-12-13

**Authors:** Seiichi Mawatari, Tsutomu Tamai, Kotaro Kumagai, Akiko Saisyoji, Kaori Muromachi, Ai Toyodome, Ohki Taniyama, Haruka Sakae, Sho Ijuin, Kazuaki Tabu, Kohei Oda, Yasunari Hiramine, Akihiro Moriuchi, Kazuhiro Sakurai, Shuji Kanmura, Akio Ido

**Affiliations:** 1Digestive and Lifestyle Diseases, Department of Human and Environmental Sciences, Kagoshima University Graduate School of Medical and Dental Sciences, Kagoshima 8908544, Japan; 2Department of Gastroenterology and Hepatology, Kagoshima City Hospital, Kagoshima 8908760, Japan; 3Department of Hepatology, Kagoshima Kouseiren Hospital, Kagoshima 8900062, Japan; 4Department of Gastroenterology, National Hospital Organization Kagoshima Medical Center, Kagoshima 8920853, Japan

**Keywords:** hepatocellular carcinoma (HCC), lenvatinib, TACE, intermediate stage, on demand

## Abstract

**Simple Summary:**

We investigated the prognosis of intermediate-stage hepatocellular carcinoma (HCC) patients who received lenvatinib (LEN) until unacceptable adverse events or progressive disease, followed by transcatheter arterial chemoembolization (TACE) on demand. The overall survival (OS) in patients for whom LEN was re-administered after TACE (TACE-LEN) was significantly longer in comparison to patients who received other therapies, such as only TACE, other drugs after TACE, or other drugs without TACE. TACE-LEN was the most associated with OS in the Cox proportional hazard analysis. In intermediate-stage HCC patients who can tolerate LEN without discontinuation due to AEs, TACE-LEN may prolong the prognosis.

**Abstract:**

The present study clarified the prognosis of intermediate-stage hepatocellular carcinoma (HCC) patients who received lenvatinib (LEN) followed by transcatheter arterial chemoembolization (TACE) on demand. We retrospectively evaluated 88 intermediate-stage HCC patients who received LEN. The median age was 74 (range: 47–92) years old, 67 patients were male, and 82 were classified as Child-Pugh A. LEN was administered until disease progression or discontinuation due to adverse events (AEs). The mean duration of LEN treatment was 7.0 months. The response and disease control rates were 51.1% and 89.8%, respectively. The median progression-free survival and overall survival (OS) after the initiation of LEN were 6.8 months and 29.9 months, respectively. The OS in patients for whom LEN was re-administered after TACE (TACE-LEN) was better than that in patients who received other therapies (e.g., only TACE, TACE-other therapy, or only other therapy) even with propensity score matching (*p* = 0.008). A Cox proportional hazard analysis showed that TACE-LEN was most strongly associated with the OS (hazard ratio: 0.083, 95% confidence interval: 0.019–0.362, *p* = 0.001). LEN was administered for approximately 11.1 months after TACE. In intermediate-stage HCC patients who can tolerate LEN without discontinuation due to AEs, TACE-LEN may prolong the prognosis.

## 1. Introduction

Liver cancer is the sixth-most commonly diagnosed cancer and the fourth leading cause of cancer death worldwide [1]. In Japan, approximately 25,000 patients died of liver cancer in 2019; this was the fifth highest rate among all cancer deaths, following deaths from colon, lung, stomach, and pancreatic cancers [2]. Hepatocellular carcinoma (HCC) accounts for most liver cancer. There are various treatments for HCC, including surgical resection, local necrosis therapy (percutaneous ethanol injection therapy: PEI, percutaneous radiofrequency ablation therapy: RFA, etc.), transcatheter arterial chemoembolization (TACE), systemic chemotherapy or liver transplantation.

The Barcelona Clinic Liver Cancer (BCLC) system is widely used for the staging of HCC, and it is classified based on factors such as the number of tumors, size, presence of vascular invasion, liver function, and performance status (PS) [3,4]. Among them, the intermediate stage (BCLC-B) is defined as Multinodular, unresectable, a Preserved liver function, PS 0, and TACE is recommended for intermediate-stage HCC [3,4]. However, TACE is not always the optimal treatment because intermediate-stage HCC is a heterogeneous disease in terms of the tumor burden and liver function; the subclassification of intermediate-stage HCC, such as the "up-to-7 criteria" are used globally to describe tumor burden [5,6].

In 2009, Sorafenib, an oral multikinase inhibitor, was approved as a first-line treatment for HCC and has been shown to improve overall survival (OS) [7]. For several years, there was no first-line treatment for HCC over Sorafenib. In 2018, Lenvatinib (LEN; Lenvima^®^ capsule) was approved for the treatment of unresectable advanced HCC after the non-inferiority of overall survival (OS) to Sorafenib was demonstrated in the REFLECT study [8]. However, the outcome of LEN treatment for BCLC-B HCC is not fully clear.

It has also been reported that combination treatment with TACE and Sorafenib improves therapeutic outcomes (TACTICS trial) [9]. Intermediate-stage HCC patients for whom TACE was expected to be insufficient have been converted to systemic therapy; however, the clinical effect after systemic therapy was not clear. We aimed to clarify the prognosis of intermediate-stage HCC patients who received LEN followed by TACE on demand.

## 2. Materials and Methods

### 2.1. Study Population

This retrospective study used medical information obtained from patients who received the most appropriate medical care for patients. Among 100 patients who were introduced to LEN for intermediate-stage HCC at our department and related facilities from May 2018 to September 2021, 88 patients who received LEN for more than 2 weeks were enrolled. A single tumor with a maximum size of ≥5 cm was classified as BCLC A [6]. The study protocol conformed to the ethical guidelines of the Declaration of Helsinki and was approved by Kagoshima University Hospital and the research ethics committee of each participating facility (approval number: 180228). An opt-out approach was used to obtain informed consent from patients, and personal information was protected during data collection.

### 2.2. Therapeutic Agent and Follow-Up

LEN (Lenvima^®^, Eisai Co., Ltd., Tokyo, Japan) was administered orally at a dose of either 12 mg/day for patients with a body weight of >60 kg or 8 mg/day for patients with a body weight of <60 kg. However, 38 patients (43.2%) started with a reduced dose at the discretion of the attending physician. Tumors were followed by dynamic computed tomography or magnetic resonance imaging at 6–12 weeks after the start of treatment according to the modified Response Evaluation Criteria in Solid Tumors (mRECIST) criteria [10]. The dosage should be reduced when a patient develops grade ≥ 3 severe adverse events (AEs) or when any unacceptable grade 2 drug-related AEs occur. Treatment continued until the development of unacceptable AEs or progressive disease. AEs were assessed using the National Cancer Institute Common Terminology Criteria for Adverse Events (CTCAE) (version 5.0) [11].

### 2.3. TACE Treatment after the Diagnosis of Progression Disease or Discontinuation Due to AEs

LEN was administered until the progression of disease (PD) or the discontinuation due to AEs. TACE was added on demand according to the condition of the tumor. LEN was re-administered at the discretion of each attending physician while monitoring the patient’s physical condition and liver function. In conventional TACE (cTACE), an emulsion containing iodized oil (Lipiodol; Guerbet, Tokyo, Japan) and miriplatin (60–120 mg; Miripla^®^; Sumitomo Dainippon Pharma, Osaka, Japan), cisplatin (50–100 mg; IAcall^®^; Nippon Kayaku, Tokyo, Japan), or Epirubicin (20–50 mg; Epirubicin^®^; Nippon Kayaku, Tokyo, Japan) were injected selectively or segmentally until high-grade stasis was reached, followed by embolization with absorbable gelatin sponge particles (Gelpart^®^; Nippon Kayaku, Tokyo, Japan). The study also included cases in which drug-eluting beaded TACE (DEB-TACE) was used instead of cTACE at the facility’s discretion. DEB-TACE was performed using DC Beads^®^ (Biocompatibles UK, Surrey, UK) impregnated with Epirubicin (50 mg).

### 2.4. Evaluation of the Liver Function

The Albumin-Bilirubin (ALBI) score and modified-ALBI (mALBI) grade were used to evaluate liver function [12].

### 2.5. Statistical Analyses

Statistical analyses were performed using the IBM Statistical Package for Social Sciences (SPSS) software program (version 22 IBM SPSS Statistics, Armonk, NY, USA). Categorical data were compared using the chi-squared test and Fisher’s exact test, as appropriate. Continuous variables were analyzed using the Mann-Whitney U test and the Kruskal-Wallis test. The Kaplan–Meier method and log-rank test were used to analyze the cumulative rates of HCC development. *p* values of <0.05 were considered to indicate statistical significance. Factors associated with the development of HCC were determined using a Cox proportional hazards analysis with forward selection using *p* < 0.10 as a cutoff for inclusion in the model.

### 2.6. Propensity Score Matching (PSM)

The propensity scores of the patients who received post-treatment, after the exclusion of patients with resection, RFA, or who received LEN as a second- or third-line treatment, were estimated using a logistic regression model with the following characteristics as covariates: age, sex, mALBI grade before therapy and at the end of treatment, number of times TACE was performed before the initiation of LEN, alpha-fetoprotein (AFP), des-gamma carboxyprothrombin (DCP), up to-seven criteria, and overall response rate (ORR). The c statistic was 0.793. A one-to-one nearest-neighbor matching algorithm with an optimal caliper of 0.2 times the standard deviation of propensity scores without replacement was used to pair the groups. Finally, 11 pairs were selected.

## 3. Results

### 3.1. Baseline Characteristics

The characteristics of 88 patients are summarized in Table 1. The median age was 74 years (range: 47–92), 67 patients were male, 82 patients were classified as Child-Pugh A, and 30 patients were classified as mALBI grade 1. LEN was administered as the first systemic therapy in 75 patients.

### 3.2. Best Response and Survival Outcomes after the Initiation of Lenvatinib

Table 2 shows the best response of patients who received LEN. The response and disease control rates were 51.1% and 89.8%, respectively. The complete response (CR), partial response (PR), stable disease (SD), and PD rates were 8.0%, 43.2%, 38.6%, and 10.2%, respectively. The Kaplan–Meier curve showed that the median progression-free survival (PFS) (Figure 1a) and overall survival (OS) (Figure 1b) after the initiation of LEN were 6.8 months (95% confidential interval (CI): 5.6–8.0 months) and 29.9 months (95% CI: 25.7–34.0 months), respectively.

### 3.3. Reasons for Discontinuing Lenvatinib

The median duration of LEN treatment was 169 days. LEN was discontinued in 85 cases (96.6%). Appendix A shows the details concerning cases of discontinuation of LEN therapy. LEN was discontinued due to disease progression in 55 (65.5%) cases, due to AEs in 29 cases (34.1%), and due to achieving CR in 1 case (1.2%). The breakdown of AEs was s follows: deterioration of liver function (n = 8), gastrointestinal symptoms including gastrointestinal bleeding (n = 8), general malaise (n = 4), respiratory disease (n = 3), and other (n = 6; including portal vein thrombus [n = 1], cerebral hemorrhage [n = 1], and thrombocytopenia [n = 1], and the exacerbation of pre-existing illness [n = 3]).

### 3.4. Post-Treatment with LEN

In 65 (76.5%) of 85 patients, LEN treatment was discontinued and switched to post-treatment. Specifically, they were divided into the following six groups: A, TACE-LEN (LEN was re-administered after TACE [n = 19]); B, TACE-other therapy (Sorafenib, regorafenib, ramucirumab, cabozantinib, or Atezolizumab + bevacizumab [ATZ + BEV] were introduced after TACE [n = 19]; C, only TACE (n = 13); D, other therapies (only drug therapy without TACE [n = 11]); E, RFA or resection (n = 3); F, no therapy (n = 20). We compared the OS and PFS rates of the six groups. The Kaplan–Meier curve for OS showed that group A had a better prognosis than groups B, C, D, or F (A vs. B, *p* = 0.002; A vs. C, *p* < 0.001; A vs. D, *p* < 0.001; A vs. E, *p* = 0.492; A vs. F, *p* < 0.001; Figure 2a). The median survival time was as follows: A, not reached; B, 28.0 months; C, 15.8 months; D, 18.6 months; E, not reached; F, 23.6 months. The Kaplan–Meier curve for the PFS showed that there was no significant difference between group A and groups B, C, D, or F (A vs. B, *p* = 0.461; A vs. C, *p* = 0.068; A vs. D, *p* = 0.139; A vs. E, *p* = 0.513; A vs. F, *p* = 0.077; Figure 2b). The median PFS was as follows: A, 6.1 months; B, 5.1 months; C, 5.5 months; D, 4.8 months; E, 16.2 months; F, 13.3 months. We compared the OS from the first discontinuation of LEN in patients who received LEN in the first line and underwent post-treatment other than resection or RFA. Similarly, the Kaplan–Meier curve for OS showed that group A had a better prognosis in comparison to group B, C, or D (A vs. B, *p* = 0.021; A vs. C, *p* < 0.001; A vs. D, *p* = 0.002; Figure 2c). The median survival time after the first discontinuation of LEN was as follows: A, not reached; B, 26.7 months; C, 8.8 months; D, 15.7 months. In addition, the median PFS in cases that received a second round of LEN therapy was 6.9 months (Figure 2d).

We investigated whether or not discontinuation due to AEs affected the PFS and OS. The Kaplan–Meier curves for the PFS and OS showed that there was no significant difference between the rate of discontinuation due to AEs and that due to PD (Appendix A). Among patients who discontinued therapy due to PD (not AEs), we compared the OS rates in the A, B, C, and D groups. The Kaplan–Meier curve for OS showed that group A had a better prognosis than groups B, C, and D (A vs. B, *p* = 0.003; A vs. C, *p* < 0.001; A vs. D, *p* = 0.001; Appendix A). Similarly, on comparing the OS from the first discontinuation of LEN, the Kaplan–Meier curve for the OS showed that group A had a better prognosis than groups B, C, and D (A vs. B, *p* = 0.009; A vs. C, *p* < 0.001; A vs. D, *p* = 0.005; Appendix A).

### 3.5. Factors Associated with the OS

The Cox proportional hazard analysis showed that the following factors were associated with OS from the introduction of LEN: TACE performed more than 2 times before the initiation of LEN (Hazard Ratio [HR] 3.747, 95% confidence interval [CI] 1.432–9.807), maximum tumor diameter > 50 mm (HR 3.035, 95% CI 1.428–6.451), number of tumors > 7 (HR 2.437, 95% CI 1.258–4.722), grade 2b or 3 mALBI at the end of treatment (HR 2.378, 95% CI 1.243–4.548), and post-treatment with TACE-LEN (HR 0.083, 95% CI 0.019–0.362) (Table 3). Similarly, the Cox proportional hazard analysis showed that the following factors were associated with OS from the first discontinuation of LEN: grade 2b or 3 mALBI at the end of treatment (HR 2.590, 95% CI 1.363–4.922) and post-treatment with TACE-LEN (HR 0.110, 95% CI 0.026–0.462) (Table 3).

### 3.6. Comparison of Background Factors According to Post-LEN Treatment

Table 4 shows a comparison of patient backgrounds by post-LEN treatment in patients who were administered LEN in the first line. Appendix A shows the characteristics of the non-TACE-LEN group. There was no difference in tumor factors (size, number, and tumor markers) and pre-treatment liver function. The administration period of LEN in the TACE-LEN group was longer than that in the non-TACE-LEN group, and the number of times TACE was administered before LEN treatment in the TACE-LEN group was lower than that in the non-TACE-LEN group, and the rate of discontinuation due to adverse effects tended to be low (*p* = 0.073). In addition, ORR in the TACE-LEN group was higher than that in the non-TACE-LEN group. In patients for whom LEN was re-administered, LEN was discontinued for 0–56 days (median 9.5 days) before TACE and re-administered 7–124 days (median 28.5 days) after TACE at the discretion of each attending physician. Ultimately, LEN was administered for an average of 331.9 days after TACE. TACE was performed approximately 2.6 times after the discontinuation of LEN. Eight cases received ATZ + BEV therapy after the discontinuation of TACE-LEN.

### 3.7. Comparison of the OS between TACE-LEN and Other Therapies after PSM

In patients who received LEN in the first line, OS was compared using PSM to adjust the background factors of the TACE-LEN and other treatment groups (non-TACE-LEN) who received post-treatment other than resection or RFA. A comparison of background factors is shown in Table 5. There was no difference in tumor factors (size, number, and tumor markers), pre- and post- treatment liver function, the number of times TACE was administered before LEN treatment, ORR, and discontinuation rates due to adverse events between the two groups. OS in the TACE-LEN group was significantly better than that in the non-TACE-LEN group (*p* = 0.008) (Figure 3a). Similarly, in the TACE-LEN group, OS from the discontinuation of LEN was significantly better than that in the non-TACE-LEN group (*p* = 0.010) (Figure 3b).

## 4. Discussion

In this study, we clarified that the patients with intermediate-stage HCC who underwent TACE after the end of LEN therapy and for whom LEN (TACE-LEN) was re-administered had a better prognosis than those who received TACE plus other therapy, TACE alone, or other therapies without TACE, although there was no difference in the pre-treatment tumor factors or liver function. In this study, the median OS was 29.9 months, which was longer than that in the REFLECT study (18.5 months) [8]. After PSM, OS was compared between the TACE-LEN and other therapy groups. The TACE-LEN group had significantly better OS in comparison to the other group. Thus far, no reports have examined treatment after the discontinuation of LEN in detail. In the TACE-LEN group, TACE tended to be performed fewer times, and the number of AEs tended to be low. Furthermore, LEN could be administered long-term after TACE. In the present study, 43.2% of patients had started dose-reduction of LEN. While paying attention to AEs, adding TACE according to the disease progression might prolong the prognosis.

There have been several reports on combination therapy of TACE and tyrosine kinase inhibitor. In the TACTICS trial, TACE plus Sorafenib significantly improved PFS over TACE alone in patients with unresectable HCC [9]. In the protocol, Sorafenib was pre-administered 2,3 weeks before TACE, and then Sorafenib was re-administered after TACE, and the endpoint was time to untreatable (unTACEable) progression. In recent years, LEN-TACE sequential therapy was reported to improve the prognosis of unresectable HCC in comparison to LEN alone, TACE alone, or TACE-SOR [13,14,15,16,17]. These reports differed from our study in the background of the HCC BCLC stage or the comparison of the treatment method (drug, TACE, etc.). Kuroda et al. reported that LEN-TACE sequential therapy might provide more clinical benefits than LEN monotherapy in patients with unresectable HCC, including BCLC-C [18]. TACE was performed after assessing the therapeutic effect at 8 weeks after LEN treatment. Our study differs in that LEN was administered until PD or discontinuation due to AEs, and TACE was added on demand. Similar to the findings described in our study, Shimose et al. showed the usefulness of LEN and trans-arterial therapy. They continued to administer LEN if the therapeutic effect of LEN was confirmed [19]. In our study, the timing of resuming LEN after TACE differed among individual cases due to factors such as side effects after TACE. In addition, we focused on post-treatment after the discontinuation of LEN. The optimal timing of TACE after the discontinuation of LEN and re-administration of LEN treatment will be issues for future studies.

Patients with intermediate-stage HCC are a heterogeneous population. Bolondi et al. reported that the results of TACE treatment were poor in cases with more than up-to-seven criteria, and such cases are considered TACE-refractory [5]. In the present study, 85.2% of the patients were outside of the up-to-seven criteria, and 78.9% of the cases in the TACE-LEN group were TACE-refractory. TACE-LEN was shown to be effective, even in cases outside of the up-to-seven criteria.

Liver function is well known to be an important factor in the prognosis of HCC. Lee et al. reported that the ALBI grade was an important factor associated with survival in patients with intermediate-stage HCC who underwent TACE [20]. In the present study, the mALBI grade at the EOT was the independent factor associated with OS from the introduction or discontinuation of LEN. It was shown that maintenance of liver function was associated with prolonged survival.

TACE is the recommended treatment for the BCLC-intermediate stage [4,21,22]. In the present study, the number of times TACE was performed before LEN treatment was also associated with the prognosis (Table 3). Kudo et al. showed a proof-of-concept study to compare the effectiveness of LEN and TACE and showed the superiority of LEN over TACE as first-line treatment for patients with intermediate-stage HCC beyond the up-to-seven criteria with Child–Pugh A liver function [23]. Similarly, Kobayashi et al. showed that LEN was efficacious in treatment-naïve BCLC B2 substage HCC patients in comparison to TACE, which was used in historical control studies [24]. The early initiation of LEN for patients with the intermediate-stage disease will lead to an improved prognosis.

Lenvatinib is a multikinase inhibitor that suppresses vascular endothelial growth factor (VEGF) receptors 1–3, fibroblast growth factor (FGF) receptors 1–4, platelet-derived growth factor (PDGF) receptor alpha, rearranged during transfection (RET), and KIT [8,25,26]. On the other hand, in tumor tissues, incomplete TACE induces ischemic conditions, leading to the upregulation of hypoxia-inducible factor 1-α (HIF-1α) [27,28]. As a result, increased HIF1-α upregulates the expression of VEGF, FGF or PDGF and increases tumor angiogenesis [27,29,30]. In addition, it has been reported that the prior administration of a VEGF inhibitor normalizes abnormal tumor blood vessels and enhances the therapeutic effect of TACE [31]. In this study, the median PFS of the second LEN therapy was similar to that of the first LEN therapy (6.9 and 6.1 months, respectively), even when first-line LEN treatment was discontinued due to disease progression or adverse events (Figure 2b,d). TACE therapy might alter the tumor microenvironment to improve the efficacy of subsequent LEN therapy.

In recent years, ATZ + BEV therapy, immune checkpoint inhibitor and VEGF inhibitor have been approved for HCC [32] and have become first-line agents for advanced hepatocellular carcinoma [4,21,22]. In this study, 25 of 85 patients who underwent post-treatment received ATZ + BEV after LEN treatment, and in the TACE-LEN group, 8 of 18 patients switched to ATZ + BEV (Table 4). No patient received ATEZO + BEV before LEN treatment in the present study. In the future, it will be necessary to examine the relationship between ATZ + BEV and LEN treatment.

Terashima et al. showed that post-progression survival influences OS among patients with advanced HCC [33]; post-treatment has become a more important factor for the prolongation of OS. In the present study, the liver function at the end of LEN treatment and TACE-LEN were prognostic factors after the discontinuation of LEN. In the TACE-LEN group, LEN was re-administered for 320 days after TACE. The administration of LEN for as long as possible and the addition of TACE may prolong survival.

This was not a randomized study, and the administration of LEN or other therapies as a post-treatment modality was decided based on each physician’s decision. We thus did not decide the treatment that would be administered after LEN discontinuation before actually delivering LEN therapy. As seen in Table 4 and Appendix A, the TACE-LEN group had a higher objective response rate than the non-TACE-LEN group. Thus, in patients with LEN response, it is possible that LEN was preferred as a post-treatment therapy following TACE therapy. In addition, the TACE-LEN group tended to have a lower discontinuation rate due to AEs than the non-TACE-LEN group. There was no marked difference in the PFS or OS between cases of AE discontinuation and non-AE (PD) discontinuation (Appendix A). In addition, similar results were obtained in patients without discontinuation due to AEs (only discontinuation due to PD; Appendix A). After PSM, the OS after LEN discontinuation in patients who received TACE-LEN was better than that in patients who received other therapies (Figure 3). However, Appendix A showed that group C (only-TACE group) tended to have a higher discontinuation rate due to gastrointestinal symptoms than other groups; therefore, LEN or other drugs might not have been re-administered in group C. Similarly, in groups B (TACE-other therapy) and D (other therapy), a few patients might have avoided LEN re-administration. We, therefore, speculate that the TACE-LEN combination helped prolong the prognosis in patients who were able to tolerate LEN administration without discontinuation due to AEs.

The present study was associated with some limitations. First, it was a retrospective study. Second, the population was relatively small. Third, a selection bias existed, even with adjustment by PSM. Fourth, in the TACE-LEN group, there were individual differences in the duration of the re-administration of LEN after TACE. Fifth, although the characteristics or backgrounds of post-treatment patients for whom LEN therapy was discontinued did not differ between groups (Appendix A), it is possible that the background, including the liver function or tumor status, might have differed after on-demand TACE therapy or at the timepoint of initiation of either the second LEN therapy or other therapy, which would influence the OS. We consider that a randomized controlled study is necessary to prove the efficacy of TACE-LEN therapy.

## 5. Conclusions

The OS of patients for whom LEN was re-administered after TACE was significantly longer in comparison to patients who received other therapies, such as only TACE, other drugs after TACE, or other drugs without TACE. In HCC patients who can tolerate LEN without discontinuation due to AEs, TACE-LEN therapy may prolong survival.

## Figures and Tables

**Figure 1 cancers-14-06139-f001:**
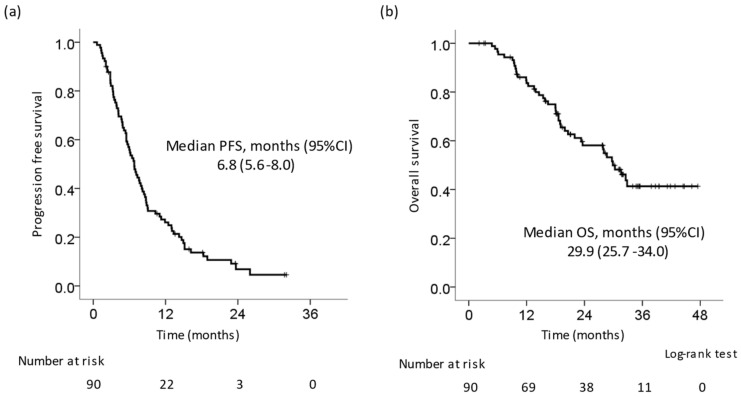
Kaplan–Meier curve of progression-free survival and overall survival in all patients. (**a**) Progression-free survival (PFS). The median PFS was 6.8 months. (**b**) Overall survival (OS). The median OS was 29.9 months.

**Figure 2 cancers-14-06139-f002:**
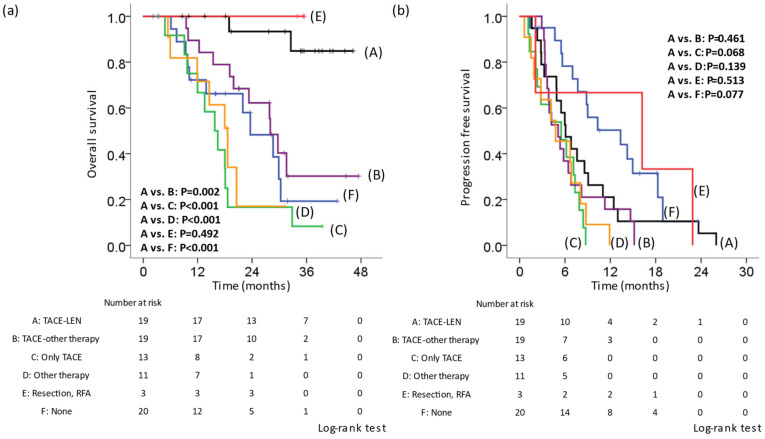
Kaplan–Meier curve of the overall survival and progression-free survival according to post-treatment in cases in which lenvatinib therapy was discontinued. Cases in which lenvatinib (LEN) was discontinued were divided into six groups. A: TACE-LEN, LEN was re-administered after TACE. B: TACE-other therapy, sorafenib, regorafenib, ramucirumab, cabozantinib, and atezolizumab + bevacizumab were introduced after TACE. C: Only TACE. D: Other therapy, only drug therapy without TACE. E: RFA or resection. F: No therapy. (**a**) OS from the initiation of LEN. The OS in group A was significantly better than that in groups B, C, D, or F. (**b**) The PFS from the first initiation of LEN. There was no significant difference between group A and groups B, C, D, or F. (**c**) OS from the first discontinuation of LEN in patients for whom LEN was administered in the first line and who underwent post-treatment. The OS in group A was significantly better than that in groups B, C, or D. (**d**) The PFS from the second initiation of LEN. LEN, lenvatinib; TACE, transcatheter arterial chemoembolization; OS, overall survival; PFS, progression-free survival.

**Figure 3 cancers-14-06139-f003:**
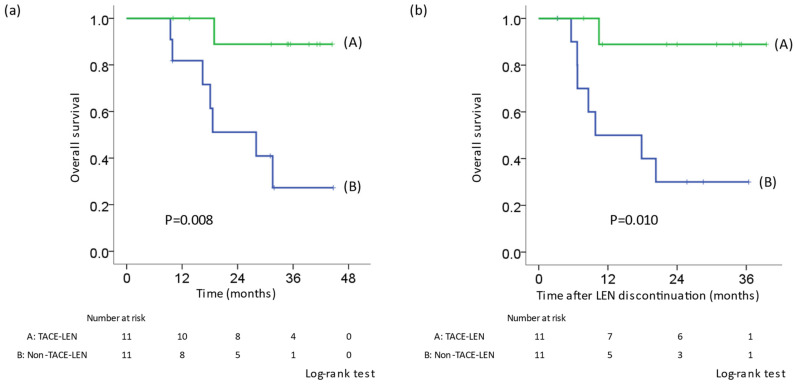
Kaplan–Meier curve for OS after adjustment by propensity score matching in patients who received LEN in the first line. TACE-LEN: LEN was re-administered after TACE. (**a**) OS from the initiation of LEN. (**b**) OS from the first discontinuation. The OS in the TACE-LEN group was significantly prolonged in comparison to patients who received other therapies. LEN, lenvatinib; TACE, transcatheter arterial chemoembolization; OS, overall survival.

**Table 1 cancers-14-06139-t001:** Patient characteristics.

Characteristics	n = 88
Age, years (range)	74 (47–92)
Sex (male/female)	67/21
Etiology (HBV/HCV/NBNC (Alcohol))	15/26/47 (10)
Child-Pugh score 5/6/7≤	59/23/6
mALBI 1/2a/2b/3	30/24/31/3
Clinical Stage II/III	11/77
Maximum tumor diameter, mm (range)	34 (10–132)
Number of tumor, 0–3/4–6/7-	18/41/29
Up-to-seven criteria in/out	16/72
Number of TACE treatments before LEN initiation, times	3 (0–12)
Systemic therapeutic line 1st/2nd/3rd	75/11/2
Platelet count, ×10^4^/µL	14.1 (5.0–42.0)
AFP, ng/mL	18.2 (2–38960)
DCP, mAU/mL	92 (12–139350)
Initiation amount, 12/8/6/4 mg	19/54/2/13
Start with a reduced amount, n (%)	38 (43.2)
Duration of LEN administration, days	169 (19–965)

Median (minimum-maximum), HBV, hepatitis B virus; HCV, hepatitis C virus; NBNC, HBV (−) and HCV (−); mALBI, modified Albumin-Bilirubin grade; TACE, transcatheter arterial chemoembolization; LEN, lenvatinib; AFP, α-fetoprotein; DCP, des-γ-carboxy prothrombin.

**Table 2 cancers-14-06139-t002:** The best response to lenvatinib, according to mRECIST.

Response	n = 88
Objective Response Rate, n (%)	45 (51.1)
Disease Control Rate, n (%)	79 (89.8)
Best response, n (%)	
Complete Response	7 (8.0)
Partial Response	38 (43.2)
Stable Disease	34 (38.6)
Progressive Disease	9 (10.2)

**Table 3 cancers-14-06139-t003:** Factors associated with the overall survival (n = 85).

Valuable	Cut off	OS from Introduction of LEN			OS from Discontinuation of LEN		
		*p* Value	Hazard Ratio	95% CI	*p* Value	Hazard Ratio	95% CI
Number of TACE treatments before LEN initiation	≥2 times	0.007	3.747	1.432–9.807			
Maximum tumor diameter	≥50 mm	0.004	3.035	1.428–6.451			
Number of tumors	≥7	0.008	2.437	1.258–4.722	0.059	1.824	0.978–3.403
mALBI at end of treatment	Grade 2b or 3	0.009	2.378	1.243–4.548	0.004	2.590	1.363–4.922
Post-treatment	LEN-TACE	0.001	0.083	0.019–0.362	0.003	0.110	0.026–0.462

Covariates were age, sex, response, lenvatinib line, and mALBI before therapy. OS, overall survival; TACE, transcatheter arterial chemoembolization; CI, confidence interval; mALBI, modified albumin-bilirubin score.

**Table 4 cancers-14-06139-t004:** Characteristics of post-treatment in patients for whom lenvatinib therapy was discontinued.

Characteristics	A: TACE-LEN n = 18	B: Non-TACE-LEN * n = 36	*p* Value
Age, years	71.0 ± 9.2	72.0 ± 9.5	0.594
Sex, Male/Female	13/5	29/7	0.358
mALBI before therapy 1/2a/2b/3	8/4/6/0	11/11/14/0	0.589
mALBI at end of treatment 1/2a/2b/3	4/5/9/0	13/11/9/3	0.211
Maximum tumor diameter, mm	44.3 ± 31.5	37.9 ± 19.9	0.847
Number of tumors 0–3/4–6/7-	3/10/5	7/16/13	0.738
Up-to-seven in/out	4/14	7/29	0.537
Number of TACE treatments before LEN initiation, times	2.3 ± 2.0	3.9 ± 3.1	0.043
Platelet count (×10^4^/µL)	12.8 ± 5.2	17.3 ± 7.3	0.018
AFP (ng/mL)	181 ± 444	176 ± 425	0.287
DCP (mAU/mL)	2263 ± 7345	467 ± 801	0.503
Duration of LEN administration, days	236.8 ± 148.1 (331.9 ± 257.6)	173.0 ± 130.4	0.015
ORR, %	61.1	30.6	0.032
DCR, %	94.4	83.3	0.500
Discontinuation due to adverse events n, (%)	11.1	33.3	0.073
Post therapy (A/B/C/D*)	18/0/0/0	0/17/9/10	<0.001
Use of atezolizumab + bevacizumab, n (%)	8 (44.4)	18 (50.0)	0.462

Mean ± standard deviation. TACE, transcatheter arterial chemoembolization; LEN, lenvatinib; mALBI, modified albumin bilirubin score; AFP, alfa-fetoprotein; DCP, des-gamma carboxyprothrombin; ORR, objective response rate; DCR, disease control rate. * including TACE-other therapy (e.g., such as Sorafenib, regorafenib, ramucirumab, cabozantinib, or atezolizumab + bevacizumab), TACE only, and only other therapy.

**Table 5 cancers-14-06139-t005:** Characteristics after propensity score matching.

Characteristics	A: TACE-LEN n = 11	B: Non-TACE-LEN * n = 11	*p* Value
Age, years	70.6 ± 9.4	70.4 ± 9.6	0.742
Sex, Male/Female	10/1	10/1	0.762
mALBI before therapy 1/2a/2b/3	3/3/5/0	5/4/2/0	0.381
mALBI at end of treatment 1/2a/2b/3	3/3/5/0	4/2/5/0	0.842
Maximum tumor diameter, mm	43.9 ± 33.2	37.2 ± 21.1	0.869
Number of tumors 0–3/4–6/7-	2/6/3	3/5/3	0.865
Up-to-seven in/out	2/9	2/9	0.707
Number of TACE treatments before LEN initiation, times	3.0 ± 2.1	2.4 ± 2.4	0.464
Platelet count (×10^4^/µL)	12.3 ± 6.0	15.8 ± 6.0	0.139
AFP (ng/mL)	262 ± 560	58 ± 136	0.178
DCP (mAU/mL)	842 ± 1818	673 ± 1171	0.670
Duration of LEN administration, days	257.4 ± 176.8	221.6 ± 92.4	0.870
ORR, %	63.6	54.5	0.500
DCR, %	90.9	100	0.500
Discontinuation due to adverse events n, (%)	1 (9.1)	1 (9.1)	0.762
Post therapy (A/B/C/D *)	11/0/0/0	0/7/2/2	<0.001
Use of atezolizumab + bevacizumab, n (%)	3 (27.3)	5 (45.5)	0.330

Mean ± standard deviation. TACE, transcatheter arterial chemoembolization; LEN, lenvatinib; mALBI, modified albumin bilirubin score; AFP, alfa-fetoprotein; DCP, des-gamma carboxyprothrombin; ORR, objective response rate; DCR, disease control rate. * including TACE-other therapy (e.g., such as Sorafenib, regorafenib, ramucirumab, cabozantinib, or atezolizumab + bevacizumab), TACE only, and only other therapy.

## Data Availability

Data contained within the article or Appendix A are available according to “MDPI Research Data Policies” at https://www.mdpi.com/ethics.

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
