# Peer review of "Clinical Effect of Lenvatinib Re-Administration after Transcatheter Arterial Chemoembolization in Patients with Intermediate Stage Hepatocellular Carcinoma"

_cancers, 2022, doi:10.3390/cancers14246139_

Round 1
Reviewer 1 Report
Summary
In this study, Mawatari et al. aimed to investigate the prognosis of intermediate-stage hepatocellular carcinoma patients who received Lenvatinib followed by transcatheter arterial chemoembolization (TACE) on demand. Although this manuscript is well described in detail, I have several issues to be discussed.
Major comments.
1. Main finding of this manuscript is the longer OS in patients treated with TACE-LEN compared to those treated with TACE-other therapy as shown in Fig. 2. The analysis using the method of propensity score matching also supported this finding. Although the characteristics or backgrounds of post-treatment in patients for whom Lenvatinib therapy was discontinued are not different among group (Table S1), it is possible that the background including the liver function or tumor status might differ after the on demand-TACE therapy, or at the timepoint of initiation of either 2nd LEN therapy or other therapy, which would influence the overall survival.
2. Patients in TACE-LEN group (Group A in Fig2) received 2nd LEN therapy after fist-line LEN treatment was discontinued due to disease progression or adverse events. Although TACE therapy could change and modify the tumor microenvironment so that following LEN therapy can work better, the data of the progression free survival in case of 2nd LEN therapy is needed.
Fig 2 only includes the comparison of overall survival, but the comparison of progression free survival is needed to understand the finding in an appropriate way.
3. This is not a randomized study, and the LEN or other therapy as a post-treatment is decided based on each physician’s decision. However, Is there any clinically setting that LEN is preferred as a post-treatment following TACE therapy?
Minor comments.
1. Several sentences are grammatically incorrect.
Line 20-21 (, [and?] TACE-LEN may prolong…)
Line 16-17 and Line 30; Both from the initiation and after the discontinuation of LEN
This phrase is not understood well.
Line 60; there are no first-line treatment for HCC [over Sorafenib??]
Line 166; LEN treatment was discontinued [and] switched…
2. The number of patients in the section of “3.4 Post-treatment with LEN” doesn’t match.
Line 166; Sixty-seven of 85 patients was discontinued [and] switched to post-treatment.
According to the following manuscript, 85 patients except for group F (n=20) received post-treatment, therefore 65, instead of 67, would be correct.
3. The description on Table 2 is missing.
Line 163-164; Best response and survival outcomes after the initiation of Lenvatinib [are shown in Table 2]
Author Response
Response to Reviewer 1 Comments
Major comments.
Point 1: Main finding of this manuscript is the longer OS in patients treated with TACE-LEN compared to those treated with TACE-other therapy as shown in Fig. 2. The analysis using the method of propensity score matching also supported this finding. Although the characteristics or backgrounds of post-treatment in patients for whom Lenvatinib therapy was discontinued are not different among group (Table S1), it is possible that the background including the liver function or tumor status might differ after the on demand-TACE therapy, or at the timepoint of initiation of either 2nd LEN therapy or other therapy, which would influence the overall survival.
Response 1: As you mention, it is possible that the background, including the liver function or tumor status, might have differed after on-demand TACE therapy or at the timepoint of initiation of either the second LEN therapy or other therapy, which would influence the overall survival.
While our data showed that there was no significant difference in the mALBI between the TACE-LEN and no TACE-LEN groups at the end of LEN treatment, we were unable to determine the background after on-demand TACE therapy or at the timepoint of the initiation of either the second LEN therapy or other therapy.
We have therefore added the following text the limitations section:
“Fifth, although the characteristics or backgrounds of post-treatment patients for whom LEN therapy was discontinued did not differ between groups (Table S1), it is possible that the background, including the liver function or tumor status, might have differed after on-demand TACE therapy or at the timepoint of initiation of either the second LEN therapy or other therapy, which would influence the OS.”
Point 2: Patients in TACE-LEN group (Group A in Fig2) received 2nd LEN therapy after fist-line LEN treatment was discontinued due to disease progression or adverse events. Although TACE therapy could change and modify the tumor microenvironment so that following LEN therapy can work better, the data of the progression free survival in case of 2nd LEN therapy is needed.
Fig 2 only includes the comparison of overall survival, but the comparison of progression free survival is needed to understand the finding in an appropriate way.
Response 2: We have now added a Kaplan-Meier curve for the PFS and the data on the PFS in patients who received a second round of LEN therapy (Figure 2). In the TACE-LEN group, the median PFS of the second LEN therapy was similar to that of the first LEN therapy (6.9 and 6.1 months, respectively), even when first-line LEN treatment was discontinued due to disease progression or adverse events. We have therefore added the following text to the Discussion section:
“In this study, the median PFS of the second LEN therapy was similar to that of the first LEN therapy (6.9 and 6.1 months, respectively), even when first-line LEN treatment was discontinued due to disease progression or adverse events (Figure 2b, 2d). TACE therapy might alter the tumor microenvironment to improve the efficacy of subsequent LEN therapy.”
Point 3: This is not a randomized study, and the LEN or other therapy as a post-treatment is decided based on each physician’s decision. However, is there any clinically setting that LEN is preferred as a post-treatment following TACE therapy?
Response 3: In the present study, we did not decide the treatment after LEN discontinuation before LEN therapy. As seen in Table 4 and Table S1, the TACE-LEN group showed a higher objective response rate and lower discontinuation rate due to adverse events (AEs) than the non-TACE-LEN group. Thus, in patients with LEN response or fewer AEs, it is possible that LEN was preferred as a post-treatment therapy following TACE therapy. However, after PSM, the OS after LEN discontinuation in patients who received TACE-LEN was better than that in patients who received other therapies (Figure 3b). We speculate that TACE-LEN combination helped prolong the prognosis.
We have therefore added the following text to the Discussion section:
“This was not a randomized study, and the administration of LEN or other therapies as a post-treatment modality was decided based on each physician’s decision. We thus did not decide the treatment that would be administered after LEN discontinuation before actually delivering LEN therapy. As seen in Table 4 and Table S1, the TACE-LEN group showed a higher objective response rate and lower discontinuation rate due to adverse events (AEs) than the non-TACE-LEN group. Thus, in patients with LEN response or fewer AEs, it is possible that LEN was preferred as a post-treatment therapy following TACE therapy. However, after PSM, the OS after LEN discontinuation in patients who received TACE-LEN was better than that in patients who received other therapies (Figure 3b). We speculate that TACE-LEN combination helped prolong the prognosis.”
Minor comments
Point 1. Several sentences are grammatically incorrect.
Line 20-21 (, [and?] TACE-LEN may prolong…)
Line 16-17 and Line 30; Both from the initiation and after the discontinuation of LEN
This phrase is not understood well.
Line 60; there are no first-line treatment for HCC [over Sorafenib??]
Line 166; LEN treatment was discontinued [and] switched…
Response 1: We apologize for the confusion and have now corrected the text.
Point 2. The number of patients in the section of “3.4 Post-treatment with LEN” doesn’t match.
Line 166; Sixty-seven of 85 patients was discontinued [and] switched to post-treatment.
According to the following manuscript, 85 patients except for group F (n=20) received post-treatment, therefore 65, instead of 67, would be correct.
Response 2: As suggested, we corrected the number of patients to 65.
Point 3. The description on Table 2 is missing.
Line 163-164; Best response and survival outcomes after the initiation of Lenvatinib [are shown in Table 2]
Response 3: We apologize for the mistake in the description and have now deleted the sentence.
Reviewer 2 Report
The OS of patients for whom LEN was re-administered after TACE was significantly longer in comparison to patients who received other therapies, such as only TACE, other drugs after TACE, or other drugs without TACE.
TACE-LEN therapy may prolong survival in patients with HCC.
The present study was associated with some limitations. First, it was a retrospective study. Second, the population was relatively small. Third, a selection bias existed, even adjustment by PSM. Fourth, in the TACE-LEN group, there were individual differences in the duration to the re-administration of LEN after TACE.
The authors consider that a randomized controlled study is necessary to prove the efficacy of TACE-LEN therapy.

Author Response
Response to Reviewer 2 Comments
The OS of patients for whom LEN was re-administered after TACE was significantly longer in comparison to patients who received other therapies, such as only TACE, other drugs after TACE, or other drugs without TACE.
TACE-LEN therapy may prolong survival in patients with HCC.
The present study was associated with some limitations. First, it was a retrospective study. Second, the population was relatively small. Third, a selection bias existed, even adjustment by PSM. Fourth, in the TACE-LEN group, there were individual differences in the duration to the re-administration of LEN after TACE.
The authors consider that a randomized controlled study is necessary to prove the efficacy of TACE-LEN therapy.
Title Is complete and very informative
Keywords are well selected and enough in total number
Reefrences ares well selected, appropriated and recently published
Response: Thank you for your comments. In future studies, we consider the randomized controlled study to prove the efficacy of TACE-LEN therapy.
Round 2
Reviewer 1 Report
Summary
In this study, Mawatari et al. aimed to investigate the prognosis of intermediate-stage hepatocellular carcinoma patients who received Lenvatinib followed by transcatheter arterial chemoembolization (TACE) on demand. The manuscript got improved in this revision, however, the manuscript still needs to be corrected.
Major comments.
1. The main point of this manuscript is whether LEN therapy after TACE would prolong the prognosis of patients. Regarding the data of median PFS, group A (TACE-LEN) had 6.1 months while group B (TACE-others) and 5.1 months as shown in line 177-180. In addition, group A had fewer adverse events (AE) than group B as described in the response 3.
These results indicate that group A, who had fewer AE, was treated with LEN for longer duration prior to TACE therapy, compared to group B. Furthermore, LEN could be re-administered in group A as a post TACE therapy, while LEN was not administered in group B, considering the risk of AE, probably resulting in longer prognosis in Group A.
Indeed TACE-LEN group had better prognosis even with the PSM, the number is too small to comprehend the results with PSM accurately.
Thus, whether patients are prone to occur AE or not, would affect the clinical results.
At present, this manuscript will make readers mislead and misunderstand that TACE-LEN would simply prolong the prognosis. The authors should comment on the factor of AE in the abstract and discussion in order to avoid the misunderstanding.
2. The text and the figure don’t match each other. In line 177-180, the results of PFS (probably “as shown in Figure 2b”) are described, however, Fig 2b in page 6 is about Overall survival. Please make sure all the texts and figures.
Author Response
Response to Reviewer 1 Comments  
Major comments.
- The main point of this manuscript is whether LEN therapy after TACE would prolong the prognosis of patients. Regarding the data of median PFS, group A (TACE-LEN) had 6.1 months while group B (TACE-others) and 5.1 months as shown in line 177-180. In addition, group A had fewer adverse events (AE) than group B as described in the response 3.
These results indicate that group A, who had fewer AE, was treated with LEN for longer duration prior to TACE therapy, compared to group B. Furthermore, LEN could be re-administered in group A as a post TACE therapy, while LEN was not administered in group B, considering the risk of AE, probably resulting in longer prognosis in Group A.
Indeed TACE-LEN group had better prognosis even with the PSM, the number is too small to comprehend the results with PSM accurately.
Thus, whether patients are prone to occur AE or not, would affect the clinical results.
At present, this manuscript will make readers mislead and misunderstand that TACE-LEN would simply prolong the prognosis. The authors should comment on the factor of AE in the abstract and discussion in order to avoid the misunderstanding.
Response 1: We investigated whether or not discontinuation due to AEs affected the PFS and OS. The Kaplan–Meier curve for the PFS and OS showed that there was no significant difference between the rate of discontinuation due to AEs and that due to PD (Figure S1), and similar results were obtained in patients without discontinuation due to AEs (only discontinuation due to PD, Figure S2).
We have now added the reason for discontinuation by the post-treatment of LEN (Table S1). Table S1 showed that group C (only TACE group) tended to have a higher discontinuation rate due to gastrointestinal symptoms. As you mentioned, LEN or other drugs might not have been able to be re-administered in group C. Similarly, in group B (TACE-other therapy) or D (other therapy), a few patients might have avoided LEN re-administration. We therefore speculate that TACE-LEN combination helped prolong the prognosis in patients who were able to tolerate LEN administration without discontinuation due to AEs.
We have now added, “In intermediate-stage HCC patients who can tolerate LEN without discontinuation due to AEs…” to the Abstract and the following text to the Discussion:
“As seen in Tables 4 and S2, the TACE-LEN group had a higher objective response rate than the non-TACE-LEN group. Thus, in patients with LEN response, it is possible that LEN was preferred as a post-treatment therapy following TACE therapy. In addition, the TACE-LEN group tended to have a lower discontinuation rate due to AEs than the non-TACE-LEN group. There was no marked difference in the PFS or OS between cases of AE discontinuation and non-AE (PD) discontinuation (Figure S1). In addition, similar results were obtained in patients without discontinuation due to AEs (only discontinuation due to PD; Figure S2). After PSM, the OS after LEN discontinuation in patients who received TACE-LEN was better than that in patients who received other therapies (Figure 3). However, Table S1 showed that group C (only-TACE group) tended to have a higher discontinuation rate due to gastrointestinal symptoms than other groups; therefore, LEN or other drugs might not have been re-administered in group C. Similarly, in groups B (TACE-other therapy) and D (other therapy), a few patients might have avoided LEN re-administration. We therefore speculate that TACE-LEN combi-nation helped prolong the prognosis in patients who were able to tolerate LEN admin-istration without discontinuation due to AEs.”
- The text and the figure don’t match each other. In line 177-180, the results of PFS (probably “as shown in Figure 2b”) are described, however, Fig 2b in page 6 is about Overall survival. Please make sure all the texts and figures.
Response 2: We apologize for the mistake in the description and have confirmed the text and figures.
Round 3
Reviewer 1 Report
This revised manuscript has addresses all of my questions and comments. Thank you for the inclusion of extra data.